# Evaluating Access to Health Care in Mothers and Caregivers of Children under Five Years of Age in Rural Communities of Yucatán, Mexico

**DOI:** 10.3390/ijerph21091243

**Published:** 2024-09-20

**Authors:** Elsa Rodríguez, Guadalupe Andueza, Ricardo Ojeda, Erin Palmisano, Louisa Ewald, Aruna M. Kamath, Abraham Flaxman, Shwetha H. Sanapoori, Bernardo Hernandez

**Affiliations:** 1Department of Social Medicine, Center of Regional Researches “Dr. Hideyo Noguchi” Biomedical Unit, Health Sciences Campus, Autonomous University of Yucatan, Calle 59 Num. 490, Col. Centro, Merida 97000, Yucatan, Mexico; 2Population Health Building/Hans Rosling Center, Institute for Health Metrics and Evaluation, University of Washington, 3980 15th Ave. NE, Seattle, WA 98195, USAbernardo.hernandez@insp.mx (B.H.); 3School of Public Health of Mexico, National Institute of Public Health of Mexico, Av. Universidad 655, Col. Santa María Ahuacatitlán, Cuernavaca 62100, Morelos, Mexico

**Keywords:** barriers, access to health care, warning signs

## Abstract

Populations in rural communities have more limited access to health care and attention than urban populations. The present study aimed to evaluate barriers to access to health care in mothers and caregivers of children under five years of age, twelve months after an educational intervention. The study was carried out from February to September 2022, and 472 mothers from eight communities in the state of Yucatán, in the southeast of the United Mexican States, participated. A comparative analysis was carried out on help-seeking times, obstacles to reaching it, and illnesses in children. The results revealed that the main barriers to access to care were long times to decide to seek help, lack of financial resources to pay for the transfer to another health unit, lack of someone to accompany the mother or caregiver when the child needed be transferred, and lack of transportation for the transfer. Disease knowledge remained at different levels in the eight communities; the significant differences occurred in four communities, one specifically for heart defects. It was concluded that, in the rural populations studied, there are barriers to access to health care which have to do with neglected social determinants, such as those related to conditions of gender, income, social support network, and the health system. Access to health care must be universal, so public health interventions should be aimed at reducing the barriers that prevent the population from demanding and using services in a timely manner.

## 1. Introduction

The population of children under five years of age presents a high risk of morbidity and mortality, mainly those under one year of age. The Sustainable Development Goals make a global call to countries to take actions that help reduce the problem of children’s health and care to reduce mortality [1]. The global mortality rate in 2021 in this age group was 38 deaths per 1000 children under five years of age [2]. Although mortality rates in children have been reduced worldwide, there are countries that to date have not met the goal set for 2030 of reducing neonatal mortality to less than 12 per 1000 registered live births (RLB) and that of children under five years of age to less than 25 deaths per 1000 RLB [3,4]. In high-income countries, deaths in children have decreased to 8 per 1000 RLB, compared to children in Mexico, where mortality is even higher, with 13 deaths per 1000 RLB, attributed mainly to neonates [2]. The causes of neonatal mortality are closely linked to the health status of the pregnant mother at the time of delivery, which can cause preterm delivery, neonatal hypoxia/trauma, and congenital anomalies. The causes in children over one year old are due to accidents and gastrointestinal and respiratory diseases [5,6]. Seeking timely help by recognizing warning signs contributes to the prevention of diseases and their complications that can lead to death in children under five years of age [5].

That is why children in this age group require special attention in terms of health care to be able to grow and develop properly [7]. We must be alert to any alteration in their health status, either physical, mental, and psychosocial, in order to obtain timely care that allows them to limit the damage or injury they present. Child health care should begin at home, and it is generally the mothers or caregivers who detect when something is not right with them [8]. The warnings that these mothers or caregivers perceive are alarm signs that lead them to seek medical help; however, the possibilities for this help to be timely are not always available. In low- and middle-income countries, the possibilities of obtaining care are reduced due to structural factors that have to do with the living conditions of children, which become barriers to access. Access to health care is a universal right of people and communities to be able to obtain the help they need when they request it with quality resources to reduce damage to health [9]. Recently, access to health care has been discussed as a complexity that requires greater attention considering all its dimensions, because the concept is multidimensional [10]. That is why access to care must be understood, not only in order to have available infrastructure but also to meet the health needs that the population demands, with financing, resources, equipment, and materials that respond to the perspectives of various social groups with adherence to the culture. Access to health must be looked at from a lens of equity to reduce gaps in access and for people to be able to enjoy quality services. Access barriers begin when the mother or caregiver, upon detecting that the child needs care, seeks help at a medical unit without obtaining it in a timely manner. The information that mothers have about how to prevent diseases in children is sometimes limited due to the same low capacity to access information about the causes that produce the disease, so the interventions carried out must be aimed at preventing the causes and barriers that limit access to care [11].

Geographic barriers have been described as barriers to care that have to do with the distances traveled by mothers and caregivers in rural areas that lengthen the time of care to reach help; the distance from health centers; the lack of resources to pay for the transfer [12]; communication barriers due to the limited information they receive about diseases, their causes and how to prevent them; and to individual factors related to poor decision-making by mothers and caregivers that translate into inequities [13]. Decision-making in the homes of mothers and caregivers of children occurs in an environment of power relations between genders. This could be observed during a study with women from Sierra Leone, in which they described power related to having some income, being financially independent, and being listened to in their social groups and by their husbands. But even with this, the male authority figure prevailed in decision-making at home, even in situations related to health decision-making [14]. This situation places her at a disadvantage, because although she knows that the sick boy or girl must be cared for, she cannot go out for help because the decision comes from the man as the authority. This power relationship between the genders, which society itself has established and in which care of children has been attributed to women, has devalued the role of men in caring for situations of illness in them [15]. These systematic differences between population groups can still be observed in low- and middle-income countries, impacting the health of children under five years of age who live in vulnerable conditions [16]. Poverty, the lack of opportunities to pursue an educational plan, and low power to decide on the care of children, as well as the lack of a decent income, are some conditioning factors for mothers and caregivers that impact the development and growth of children.

Until 2018, the Mexican health system was segmented and organized according to three aspects: the population with institutional health insurance, the uninsured, and those requesting a private service. In the first case, the financing is tripartite, that is, the contributions are provided in part by the company where the worker works, another by the worker, and another by the health institution to which he or she belongs. In the second case, the institution to which the uninsured go belongs to the Ministry of Health; and in the last case, there is the population that has the economic resources to finance the medical service privately. The health institutions that provide services to the insured population are the Mexican Social Security Institute (IMSS), with the greatest coverage (33%), and the Institute of Social Security and Services of State Workers (ISSSTE), with coverage of the 7.4%. However, despite being affiliated with an institution, there is a large percentage of the population that seeks care outside their institution due to access barriers and the search for higher quality services. In addition to the IMSS and ISSSTE, other insured employees who work for the government are part of the Health System, such as those of Petróleos Mexicanos (PEMEX), the military in the Secretariat of National Defense (Sedena), and sailors in the Secretariat of the Navy (Semar). The Ministry of Health is responsible for covering the uninsured population with public health and primary care programs with hospitalization, often with limited resources to provide quality care [17]. In response to this great demand for coverage of the uninsured population, Seguro Popular was created, which offered a package of services that brought help to the population, with coverage of 43.5%. Currently, with the change of Mexican government, the organization of the health system is being reorganized; it is mentioned in official narratives that the health system is universal, that is, that there is no longer segmentation of services and that everyone has the service and medications when they request them in any health institution, which is provided through of the National Institute of Health for Welfare (INSABI), which replaced Seguro Popular. The INSABI is a decentralized body of the Federal Public Administration, with its own legal personality and assets, sectored in the Ministry of Health [18]. The transition that began with the new government, in the reorganization of the health system in Mexico, had drastic repercussions on the mortality of the population, because starting in 2020, the COVID-19 pandemic occurred, coinciding with the change in the network of providers, shortages of resources and medicines, and the occurrence of excess deaths of Mexicans [19], which reflected the lack of organization and ethics in the application of rules and laws of the administration without respect for the human rights and dignity of sick people.

The Mexican Republic is divided into 32 territorial zones called States, each of which is subdivided into Municipalities that, depending on the number of inhabitants, have several cities, villages, and small communities. The State of Yucatán is in the southeast of the Mexican Republic and reported in 2020 a population of 2,320,898 inhabitants, of which 1,180,619 were women and 1,140,279 were men; it occupies 22nd place out of 32 federal entities nationwide for its number of inhabitants. Of its population, 86% is urban and 14% rural. The State has 106 municipalities and its capital is the city of Mérida. The average schooling of the population aged 15 and over is 9.5 years; 24% speak the Mayan language [20]. The population of children under five years of age for 2020 in Yucatán was 175,986, of which 89,335 were boys and 86,651 girls [21].

In 2022, 19,612 deaths in children under 1 year of age were recorded in Mexico, of which 297 (1.51%) occurred in Yucatán; of these, 115 (38.7%) occurred in the early neonatal period, 60 (20.2%) in the neonatal period, and 122 (41.0%) were postnatal deaths [22]. Regarding the deaths of children between 1 and 4 years old, in that same year, 58 deaths were reported, with the rate of deaths due to respiratory diseases being 19.3 per 100,000 and the rate of diarrheal diseases being 11.9 per 100,000 [23].

In the State of Yucatán, Mexico, the administrative distribution of health centers, clinics, and hospitals throughout this territory, intended for the care of the population, was carried out more than 40 years ago, being divided by Health Jurisdictions I, II, and III. Jurisdiction I is responsible for providing coverage to the population of the municipalities in the northern and western areas of the State, Sanitary Jurisdiction II to those in the Eastern zone, and Sanitary Jurisdiction III to those in the Central-South area. In each Jurisdiction, there are health centers and clinics that are linked to second- and third-level hospitals, the latter concentrated in the central area of the Yucatecan territory. Recent reports refer to a ratio of 2.30 doctors in public institutions in contact with the patient per 1000 inhabitants [24]. With this panorama, the scenario that the population is experiencing becomes obvious, which includes mothers and caregivers of children under five years of age when they leave their homes in search of medical help when their children get sick.

In the State of Yucatán, Mexico, there are areas with a higher risk of mortality in children under five years of age, such as the municipalities of Bokobá, Tekantó, Buctzotz, Calotmul, Cantamayec, Sotua, Chapab, and Dzan. Therefore, this study aimed to evaluate barriers to access to health care in mothers and caregivers of children under five years of age twelve months after an educational intervention.

## 2. Materials and Methods

The present study is an analytic cross-sectional design (the parameters were measured only one-time) and gives follow-up to an educative intervention with pre–post design, finalized a year before (Stage I). In the present study, we utilized the same questionnaire, pre–post administered to the mothers and caregivers of children under five years old, as in the previous intervention. After analysis of Stage I, control communities also received the educative intervention for ethical considerations.

### 2.1. Populations and Site

The participating population was mothers and caregivers of children under five years of age that had participated in the pre–post intervention (Stage I) and were citizen from 8 involved communities: Bokobá, Tekantó, Buctzotz, Calotmul, Cantamayec, Sotua, Chapab, and Dzan. The communities that received the intervention in Stage I (intervention communities) were Bokobá, Buctzotz, Cantamayec, and Chapab; those that did not (control communities) were Tekantó, Calotmul, Sotuta, and Dzan.

### 2.2. Sample

The sample was composed of 472 mothers and caregivers. The intention of the present study was to have the participation of all or the greatest number of the group of women participating in Stage I.

### 2.3. Description of the Previous Community Intervention

The previous intervention lasted one year and six months. Work teams were formed that included researchers, doctors, social workers, and health promoters who knew the communities and kept records of the participating mothers and caregivers.

Researchers from the Institute of Health Evaluation and Metrics (IHME) at the University of Washington and the Autonomous University of Yucatán (UADY) planned the questionnaire, the variables, and the intervention analysis plan. The project began with a detailed review of the interventions in the study areas, which could contribute to the development of intervention and evaluation plans. In that Stage I, a house-to-house census was carried out in each community to detect the addresses of caregivers and mothers of children under five years of age. The questionnaire was piloted on a small sample of mothers to clarify the questions and make them relevant to answer the study variables. With the support of geographic areas of basic statistics (AGEBS in Spanish), which include maps of each municipality in the State of Yucatán with its number of blocks [25], 50 homes of children under five years of age in each community were randomly selected. For that, we divided each community into five sectors, so all children were represented in the North, South, East, West, and Center of the community. Intervention and control municipalities were also previously randomized. A pre-test was applied on seeking care in general and on knowledge of warning signs for seeking care for respiratory and gastrointestinal diseases and heart defects. In this experience, it was possible to obtain the baseline state of knowledge (pretest) of the community about warning signs and the times of seeking care. After that, educational materials on these same topics were distributed to mothers and caregivers of municipalities selected as group of intervention; approximately two weeks were given to review them, and the same survey was applied again (post). The time for reviewing the educational materials was agreed upon with the mothers and caregivers, who considered it a reasonable time to become familiar with the information provided through a booklet on the topics. After this time and prior scheduling, the interviewers returned to the homes and the same pre-questionnaire was applied to the participating mothers and caregivers. After the post-test, the analysis of the results was carried out, measuring the effect of the intervention, comparing the pre-test and post-test means of knowledge of the variables studied, and measuring the impact with difference-in-differences method. Hypothesis tests were applied and the results were interpreted as significant when the *p* value < 0.05. After the statistics analysis, the control communities also received the materials with the information as an ethical consideration.

### 2.4. Description of the Current Community Assessment

In the current study, a local work team was organized again. Researchers, promoters, and social workers of UADY visited the mothers and caregivers in the 8 communities, and the same questionnaire on knowledge of alarm signs, seeking care, and access to health care was applied.

Information obtained was captured in a database. The barriers to access to care for mothers and caregivers of children under five years of age were analyzed, so as to ascertain the state of knowledge about the warning signs for the main causes of illness in children which trigger a search for care. Initially, the municipal authorities of each of the study communities were contacted; when there was a new administration, they were informed about the intervention carried out in Stage I, as well as the purposes of the current study, and to request technical support from the community health promoters who also participated in Stage I. In the current evaluation, with the support of these personnel, who had registered addresses of mothers and caregivers, contacted them again. After informed consent was signed, the same survey from the intervention carried out 12 months before was applied to the mothers in their homes to explore the current state of knowledge about the main warning signs for gastroenteritis, respiratory diseases (influenza), and congenital heart defects in children under five years of age, as well as the times of seeking care and barriers encountered during the search for help. Since the Mayan language is also spoken in the communities in addition to Spanish, the community promoters helped with translation when the case required it. Informed consent was signed before. The protocol was reviewed and authorized by Ethical Committee of Autonomous University of Yucatán, México. Code: CEI-01-2022, approved 26 January 2022.

### 2.5. Community Evaluation of the Current Study

The community evaluations of the present study consisted of comparing means and percentages of barriers to seeking care and levels of knowledge for the warning signs, which were achieved in a previous community intervention (Stage I) and during the current study. Statistical analyses were carried out globally and grouping observations at the community level. Means of knowledge obtained in the last measurement of the previous intervention (Stage I) served as a reference to compare with mains of current study, which gave a follow-up to knowledge on help-seeking times, barriers to reach health care, and alarm signs of illnesses in children. Means and standard deviations of knowledge were calculated and compared. Confidence intervals to 95% for the difference in means were calculated. Associations of barriers with care-seeking times were analyzed with odds ratio calculated. Epi-Info version 7 statistical programs and digital calculators for comparison of means and percentages with calculations of confidence intervals were used.

All activities carried out in previous and current study are described in Figure 1.

## 3. Results

Data were collected in 486 of the 492 women and caregivers of children under five years of age in Stage I. Of these, the information was complete in 472 (97.1%). Of the 472 participants, 468 (99.15%) were women and 4 (0.85%) were men.

### 3.1. Description of Population Studied

#### 3.1.1. Age of Mothers and Caregivers

The age range was from 19 to 87 years, and the highest frequency of participants (79%) was found in the groups between 20 and 39 years (Table 1).

#### 3.1.2. Other Sociodemographic Characteristics

Of the 472 participants, 351 (74.3%) had completed basic education and 121 (25.7%) had not completed primary or secondary school; 175 (37.1%) spoke the Mayan language and Spanish; and 297 (62.9%) spoke the Spanish language and understood the Mayan language but did not speak it.

#### 3.1.3. Homogeneity Analysis

To analyze the homogeneity between the groups by municipality, confidence intervals were calculated for age for each participating municipality, finding a significant difference only between the participants from the municipalities of Cantamayec and Sotuta; the highest percentage of participants also corresponded to these municipalities and those of Chapab and Dzan (Table 2).

#### 3.1.4. Levels of Knowledge of Warning Signs by Municipality and Disease

When analyzing the level of knowledge that mothers and caregivers have about the warning signs for respiratory diseases, gastrointestinal diseases, and heart defects, it was generally found that knowledge remained above 70% in the eight municipalities. In Sotuta and Chapab, knowledge also improved when compared to their reference municipality; even more so in Calotmul, which reached 100% for respiratory disease. However, significant differences occurred between the municipalities of Cantamayec vs. Sotuta and Chapab vs. Dzan. In the case of the municipality of Dzan, specifically for heart defects, knowledge was lower compared to its reference municipality (Table 3).

#### 3.1.5. Comparing Previous vs. Current Percentages on Knowledge of Warning Signs

When comparing the percentage of knowledge obtained in women and caregivers in the last intervention 12 months ago vs. current knowledge, it was observed that the recognition of two or more warning signs for gastrointestinal diseases and the recognition of general warning signs were significantly reduced. However, knowledge increased significantly for heart defects (Table 4).

#### 3.1.6. Difference of Means for Seeking Care and Warning Signs

When comparing the means of seeking care and the means of recognizing warning signs of illness obtained after the intervention 12 months ago with the means after collecting the current data, the means of current knowledge were lower than those of the previous intervention, although these differences were not significant. The greatest difference in means found was for the recognition warning signs for heart defects (Table 5).

#### 3.1.7. Comparing Care-Seeking Times Previous and Current

When comparing the times it took mothers and caregivers to seek medical attention when they recognized warning signs, it was observed that 213 (45%) in the last six months their children had become ill; and of these, 111 (52.1%) took more than 24 h to seek care. When comparing these results with those of the previous stage, differences of 1.4% greater percentage points were obtained that were not significant (*p* = 0.3571) (Table 6).

### 3.2. Analysis of Barriers

#### 3.2.1. Main Barriers

Regarding other types of barriers that mothers and caregivers faced when the child became ill, it was observed that the most frequent were not having enough money for transportation and care, and second was the lack of a companion (Table 7). However, there was no relation between lack of transportation and not having money (OR = 0.02, IC 0.0056, 0.0956).

#### 3.2.2. Transportation

Of 213 children who became ill, 148 mothers and caregivers required a means of transportation to transport their sick children to be treated, but only 93 (62.8%) had a vehicle for the transfer. The municipalities that reported the highest percentage of difficulty in obtaining transportation at the time of the illness were Sotuta and Cantamayec (68.2% and 50.0%, respectively) (Table 8).

#### 3.2.3. Associating Barriers with Time to Seeking Care

When performing the analysis of the association of barriers with the time spent seeking help, significant associations were found such as not having money for transportation and not having a companion during the transportation (OR = 65.73 IC 27.4151–157.6090 x^2^ = 124.8167 *p* = 0.0000 and OR = 15.96 IC 6.7999–37.4895 x^2^ = 52.73 *p* = 0.0000, respectively); it was also found to be associated with not having someone to leave their children with, which was not significant (OR = 1.93 IC 0.6997–5.3752 x^2^ = 1.09 *p* = 0.2952) (Table 9).

## 4. Discussion

This study demonstrates that, through community interventions aimed at mothers and caregivers of children under five years of age, knowledge about warning signs for the most common diseases in children can present changes in levels 12 months after the intervention has been carried out. The methodology used contributes to having more robust evidence on the effects of the intervention, which provided reliable results that could be used to apply the strategies to all at-risk municipalities in the state of Yucatán. Therefore, extending the intervention to other municipalities will be of utmost importance in the future in order to acquire more evidence that will allow us to address this public health problem that prevails mainly in rural areas, as morbidity is frequent in vulnerable areas of the State of Yucatán, Mexico.

In the eight municipalities evaluated, it was found that knowledge remained, with levels between 70% and 100%. The benefit of this type of intervention has also been demonstrated in other Latin American countries, mainly for diarrheal and respiratory diseases; Although mothers and caregivers were able to recognize warning signs, they were not always able to realize the severity of the disease in children, which delayed timely care and treatment [26,27,28,29]. In the eight municipalities evaluated, in Stage I, mothers and caregivers were provided with educational material through an illustrated booklet. When, in the present study, the homes were visited to carry out the survey, it was found that these materials were within the reach of mothers and caregivers, so this may have helped them to review and maintain knowledge. However, although general knowledge of warning signs still prevailed, there were differences between the percentage reached 12 months ago and current knowledge, as was the case of warning signs for gastroenteritis and for seeking care, in which they reduced the percentage significantly. The opposite was true for heart defects, where knowledge increased significantly in the current study. Regarding heart defects, it is striking that there are no articles in the literature that promote community prevention for these disorders. There are publications on interventions carried out to prevent folate and multivitamin deficiency [30,31], but we did not find reports of community interventions aimed at health promotion on heart diseases in childhood.

In Mexico, the latest reports of interventions in caregivers and family members of children under five years of age regarding respiratory and diarrheal diseases, before the intervention 12 months ago, date from 2013, which we consider to be the most recent; 20 years after they were carried out, they demonstrate the forgetfulness of this type of intervention in communities where the main morbidity in this age group is diarrheal and respiratory diseases [32,33]. Regarding the times of care for children when they became ill, it is important to mention that obtaining a vehicle for transportation continues to be the main barrier that causes delays in care. However, even though this barrier delays care, mothers and caregivers find a way to get a vehicle, although the lack of money to pay for it was not a limitation.

In the communities studied, the organization of health services is made up of local health centers, which are in the municipal capitals and in some smaller communities belonging to each municipality. In health centers, care is provided by the general practitioner. According to population coverage indicators, health centers located in municipal capitals have more staff than smaller health centers. In most of the latter, the service is provided only by a social service practitioner, a future physician close to graduating sent to the communities during his last year of study. The health services offered are those corresponding to the first level of care, that is, care for mild cases of illness and vaccination, with a basic set of medications which is scarce, and limited resources and equipment to deal with emergency situations. At this level, there is also a private physician and people who practice traditional medicine (herbalism). When the case cannot be resolved at this level, patients are referred to a second-level clinic or hospital, which corresponds to the patient according to the area of influence designated by the State Ministry of Health. Depending on the area and the communication routes available, the distances traveled in the transfer from the health center to the clinic are one to three hours if transportation is available, but this time can be extended if not. If, upon arrival at the second-level clinic or hospital, there is no resolution capacity yet due to the seriousness of the patient or due to lack of resources and equipment, the child must again be referred to another public hospital located in the state capital, where specialized services are provided. These services are concentrated in one hospital for the uninsured population, 4fourfor the population that has health insurance, and one highly specialized hospital where any type of population, whether insured or not, can go. It is under this panorama that the barriers to access to health care become evident, which prevail in the population of the eight rural communities that participated in this study.

This contrasts with other countries, where the barriers that prevail are mainly due to the delay in decision-making by mothers and caregivers [34]. In other studies, carried out in Yucatán, the need to reorganize the health system has been demonstrated because regionalization was carried out in the 80’s and sociodemographic conditions have changed, the population has increased, and land communication routes have diversified. There is a need to study more about the needs of mothers and caregivers when they face barriers to seeking help when their children get sick. Several of these barriers are highlighted in our study and in others carried out in Yucatán, some of them being the difficulties of economic resources and of obtaining a vehicle to transport the communities to clinics or hospitals far from them, inequalities that are still neglected in the state of Yucatán [35,36].

Since both the Mayan language and Spanish are spoken in the communities, the presence of the translator is very important so that there is good communication between the transmitter and the receiver. In the study, all the mothers and caregivers who participated understood and spoke Spanish, but the translator was able to clarify some doubts about the survey and informed consent in the Mayan language.

It will be of utmost importance in the future to continue extending the intervention to municipalities in the state of Yucatán, to collect evidence that strengthens decision-making about the benefits of action research, and to reach more caregivers and mothers of children under five years of age to prevent illnesses in their children.

## 5. Limitations

In the community-level evaluation, there were limitations in the number of participants between the original sample and the final sample in the current evaluation because not all the people who had participated in the intervention 12 months ago could be located due to change of domicile to another municipality. Also, we could not have a control group as in the previous study (Stage 1) because the entire sample of participants had received the intervention. Likewise, the heterogeneity in some age groups between the samples could influence some results when comparing the averages of knowledge achieved about the alarm signs that the mothers and caregivers recognized. However, in quasi-experimental preview study (Stage 1) and in current analytic cross-sectional, we attempted to reduce this limitation with the statistical analysis carried out, adjusting the knowledge means by age. Other limitations were that we were not able to obtain knowledge about warning signs for COVID-19, although educational material on warning signs and preventive measures was distributed through a printed booklet on the subject. Likewise, although some information was collected on emergency plans constructed by mothers for the care of their children, it is a task that remains to be completed in future studies.

## 6. Conclusions

This study contributed with evidence that demonstrates that, at a global level, the knowledge of mothers and caregivers about warning signs for seeking care, gastrointestinal disease, respiratory disease, and heart defects remains 12 months after the last intervention since more than 80% of mothers and caregivers manage to respond to at least two alarm signs; however, in some municipalities, there were reductions in knowledge. Regarding congenital heart defects, there were improvements, but there are still gaps in the knowledge of conditions of this type in the population studied. It is necessary to continue the evaluations of the intervention and be able to reinforce preventive strategies in the municipalities when the evidence demands it. Likewise, it is necessary to address areas of opportunity for improvement, such as the recognition of warning signs for heart defects, which is one of the main causes of mortality in neonates and children under five years of age. This study concluded that in rural populations, the barriers to access to care that prevail when mothers and caregivers seek help for childcare are related to gender conditions, which have to do with the time it takes mothers and caregivers when deciding to seek help, with socioeconomic factors such as lack of money to pay for transportation, support from the family network, and health promotion to prevent the main childhood diseases. Access to health should be universal, so public health interventions should focus on reducing the barriers that prevent the population from receiving timely care when they seek help.

## Figures and Tables

**Figure 1 ijerph-21-01243-f001:**
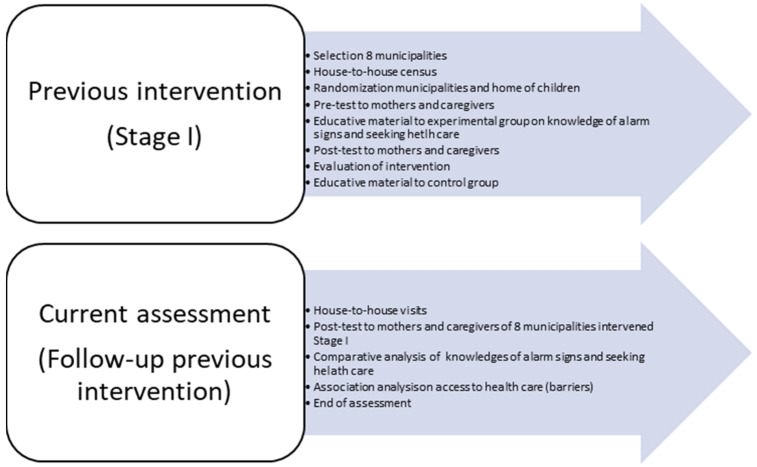
Summary of activities developed during previous intervention (stage I) and current assessment.

**Table 1 ijerph-21-01243-t001:** Mothers and caregivers of children under five years of age by age group n = 472.

Age Group	Number	%
20–24	64	13.5
25–29	112	23.7
30–34	116	24.6
35–39	81	17.1
40–44	40	8.5
45–49	19	4.0
≥50	39	8.4
Total	471	99.8

**Table 2 ijerph-21-01243-t002:** Comparison of mean ages to analyze homogeneity between the groups of mothers and caregivers participating in the study. n = 472.

Municipality	Number	%	Mode of Age	Mean of Age	CI 95%
Bokobá	45	9.3	24	31.9 ± 8.01	
Tekantó	46	9.8	27	32.7 ± 9.55	−4.0431, 3.2431
Buctzotz	55	11.8	25	34.2 ± 12.29	
Calotmul	55	11.8	34	35.31 ± 12.24	−5.4373, 3.2173
Cantamayec	71	15.0	24	33.08 ± 9.54	
Sotuta	67	14.2	37	36.27 ± 8.94	−8.1934, 2.1466
Chapab	59	12.5	31	35.09 ± 9.31	
Dzan	74	15.6	29	33.36 ± 10.70	−1.4125, 4.8725

**Table 3 ijerph-21-01243-t003:** Current level of knowledge for warning signs of respiratory, gastrointestinal disease, and heart defects between intervened municipalities (Stage I).

Municipality/Sample	Respiratory Disease		Gastrointestinal Disease		Hearth Defects	
	%	*p*=	%	*p*=	%	*p*=
Bokobá/45	93.0		95.5		77.2	
Tekantó/48	89.6	0.5019	87.5	0.1807	70.8	0.4452
Buctzotz/57	98.2		98.2		92.8	
Calotmul/57	100	0.2832	94.7	0.2758	89.4	0.5849
Cantamayec/73	43.0		57.3		9.5	
Sotuta/69	63.6	0.0170	85.1	0.0002	25.4	0.0107
Chapab/61	57.3		60.6		31.1	
Dzan/76	27.6	0.0003	42.1	0.0362	18.4	0.0757

**Table 4 ijerph-21-01243-t004:** Comparison of previous and current percentages of women who recognized two or more warning signs.

Variable	% Previous n = 492	% Current n = 472	Difference %	Z-Score	*p*-Value
Seeking medical care	85.0	88.0	3.0	−1.0448	0.2960
Influenza	67.0	68.0	1.0	−0.9908	0.3217
Gastrointestinal	88.0	74.0	14.0	3.9430	0.0000
Hearth defects	24.0	47.0	23.0	−5.4662	0.0000
General warning signs	85.0	69.0	16.0	4.2331	0.0000

**Table 5 ijerph-21-01243-t005:** Analysis of means of knowledge of seeking care and warning signs after the intervention 12 months ago and current knowledge.

	Previous Mean	Current Mean	CI	Difference of Means
Seeking medical care	3.64	3.17	0.37, 5.97	0.47
Influenza	3.28	2.78	0.31, 4.01	0.5
Gastrointestinal	3.28	3.11	0.85, 5.36	0.17
Hearth defects	2.99	2.16	0.72, 3.60	0.83

**Table 6 ijerph-21-01243-t006:** Comparison of care-seeking times in mothers and caregivers of children under five years of age who became ill.

	Previous n = 138		Current n = 213	
	Ill	%	Ill	%
˂24 h	68	49.3	102	47.9
≥24 h	70	50.7	111	52.1

*p* = 0.3571.

**Table 7 ijerph-21-01243-t007:** Main barriers faced by mothers and caregivers when transferring for childcare. n = 213.

Type of Barrier	Number of Mothers and Caregivers	%
Lacks money	117	45.5
Mothers lack a companion	67	26.1
Lack of transportation	55	21.4
Mothers have no one to leave their child with	18	7.0

Total of barriers faced: 257 *. * More than one barrier per mother/caregiver.

**Table 8 ijerph-21-01243-t008:** Percentage of women by place of residence who had transportation for the transfer. n = 148.

Residence	Number		%
Bokobá n = 15	Yes	12	80.0
	No	3	20.0
Tekantó n = 23	Yes	19	82.6
	No	4	17.4
Buctzotz n = 28	Yes	19	67.8
	No	9	32.2
Calotmul n = 23	Yes	16	69.5
	No	7	30.5
Cantamayec n = 22	Yes	11	50.0
	No	11	50.0
Sotuta n = 22	Yes	7	31.8
	No	15	68.2
Chapab n = 7	Yes	4	57.1
	No	3	42.9
Dzan n = 8	Yes	5	62.5
	No	3	37.5

Total = yes 93 (62.8%), no 55 (37.2%).

**Table 9 ijerph-21-01243-t009:** Analysis of association of access barriers with time seeking care and adjusted OR.

Barrier	≥24 h	<24 h	OR	IC	$x^{2}$	*p*-Value
Lack of money			65.73	27.4151, 157.6090	124.81	0.0000
Not	102	15				
Yes	9	87				
Lack a companion			15.96	6.7999, 37.4895	52.73	0.0000
Not	60	7				
Yes	51	95				
Lack of transportation			0.93	0.4687, 1.8574	0.0005	0.9829
Not	34	21				
Yes	59	34				
Lack of caregiver			1.93	0.6997, 5.3752	1.0926	0.2958
Not	12	6				
Yes	99	96				
Adjusted OR: 6.0609						
CI: 4.2181, 8.7089						
$x^{2}$ = 123.7899						
*p* = 0.0000						

## Data Availability

All data generated in this study are under the protection of the authors. If anyone requires any information, please write to the corresponding author or principal author.

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
