# Peer review of "Evaluating Access to Health Care in Mothers and Caregivers of Children under Five Years of Age in Rural Communities of Yucatán, Mexico"

_ijerph, 2024, doi:10.3390/ijerph21091243_

Round 1
Reviewer 1 Report
Comments and Suggestions for Authors
Dear authors:
- It's not clear the definition or concept of access that you follow. Healthcare access is usually multidimensional;
- It's necessary to further explain your methodology. It should be explained that it's a follow up of a previous intervention/study, but the reader as to understand the methodology and the instruments used and also the variables that were studied. I suggest a major revision of this proposal.
Best regards
Comments on the Quality of English LanguageExtensive editing of English language required
Author Response
It's not clear the definition or concept of access that you follow. Healthcare access is usually multidimensional;
We have added and clarified the concept of access
- It's necessary to further explain your methodology. It should be explained that it's a follow up of a previous intervention/study, but the reader as to understand the methodology and the instruments used and also the variables that were studied. I suggest a major revision of this proposal.
- We gave major clarify to the methodology, explaining that it´s a follow-up study that follow a previous.
Thank you very much for your review and comments.
Reviewer 2 Report
Comments and Suggestions for Authors
The authors have tried to assess the effect of interventions such as educating caregivers and mothers to access help/healthcare when their children require attention or medical help.
This study does not seem to have been thoroughly done or explained.
It is not clear what kind of interventions and how they were done. The assessement/evaluation of the intervention is not provided or clear.
In the initial study they are assessing the effect of education/awareness after 2 weeks. This is rather unreasonable.
The logic behind this study and the observed results are not clear.
Stage I, stage II, are all mentioned but it is unclear what the objectives of each of these were and how it was measured/evaluated.
The results appear very inconsequential and insignificant.
The study design is not well thought out. The paper also needs to be written in a better way to ensure the message and information is conveyed appropriately.
Comments on the Quality of English Language
The results appear very inconsequential and insignificant.
This may be because there is need for extensive improvement in writing and presentation, as important information or the message that authors are trying to convey is lost due to expression. It is not succinct and clear. o
Author Response
This study does not seem to have been thoroughly done or explained. We explain the study better in the methodology.
It is not clear what kind of interventions and how they were done. The assessement/evaluation of the intervention is not provided or clear. We describe in methodology that this a follow up study. We have included a figure with activities carried out in previous and current study.
In the initial study they are assessing the effect of education/awareness after 2 weeks. This is rather unreasonable. The time for reviewing the educational materials was agreed upon with the mothers and caregivers, who considered a reasonable time to become familiar with the information provided through a booklet on the topics. This text was included in the article.
The logic behind this study and the observed results are not clear. This is a follow up study that follow a previous intervention.
Stage I, stage II, are all mentioned but it is unclear what the objectives of each of these were and how it was measured/evaluated.
We extended the description of the study. In methodology we are mentioning that the present study is a follow-up of previous intervention. We only measured once in this study. In the previous study, the variables were measured twice, before and after. We used the mean after to compare with the mean obtained in current study. We wanted to know the current state of knowledge.
The results appear very inconsequential and insignificant.
We think is important to know if the effect of previous intervention persisted, to extend the intervention to other areas of the State of Yucatan.
The study design is not well thought out. The paper also needs to be written in a better way to ensure the message and information is conveyed appropriately.
In Methodology we have greater clarity to the description in the previous and current intervention.
In the initial study they are assessing the effect of education/awareness after 2 weeks. This is rather unreasonable.
The time for reviewing the educational materials was agreed upon with the mothers and caregivers, who considered a reasonable time to become familiar with the information provided through a booklet on the topics. We have included in the text of document.
The logic behind this study and the observed results are not clear.
We clarify in methodology that this is a study that follows up on a previous one. We mentioned the variables studied and the results obtained.
Stage I, stage II, are all mentioned but it is unclear what the objectives of each of these were and how it was measured/evaluated.
In Methodology we describe the objectives of each studies: previous and current.
The results appear very inconsequential and insignificant.
We think that the results are consistent and importants, because they provide whether or not the effect of the intervention remains over time.
The study design is not well thought out. The paper also needs to be written in a better way to ensure the message and information is conveyed appropriately.
To improve the message that we want to transfer to the reader, we have analyzed the comments and suggestions poin by point, trying to clarify and enrich the information in the writing, in each of the sections, with enphasis of details of the methodology.
Thank you for your valious comments and suggestions.
Reviewer 3 Report
Comments and Suggestions for Authors
The manuscript presents the results of a pre-post evaluation of an educational intervention with mothers and caregivers of children under five years of age in eight communities in Yucatán, Mexico: four communities where the intervention was implemented and four controls. The study aimed to evaluate barriers to access to health care after 12 months of the educational intervention. Its content contributes with evidence to improve health literacy and public health interventions to reduce the barriers to health care access.
It´s an interesting study, but it should be improved in some aspects before publication, especially in the methods and results description, as commented in the following.
1) Abstract
The abstract is too extended (329 words) and not objective. Authors can reduce the introduction to the theme and focus on showing the objective, methods, main findings, and conclusion.
2) Introduction
In the introduction, the authors show the differences between countries in the mortality rate of children under 5 years of age and then present the need for special attention in this group's health care, the geographic barriers to access health care, and explain the territorial health jurisdictions in Mexico. Although they summarize the function of health jurisdiction, it is essential to explain briefly how Mexico's Health System is organized and the specificities of Yucatán, if they exist, so that the reader from abroad can understand the study context.
3) Materials and Methods
This section should be more precise and objective. Methods description has a lot of repetitions. Examples of repetition:
“The community sample was represented by the same women who participated in 107 Stage 1, which was an average of 50 women in each municipality.“ (l. 107-108)
“Baseline information was captured through a survey of 50 mothers of children 130 under five years of age in each of the eight randomly selected communities, for a total of 131 400 women.“ (l. 130-131)
The design of the study can be clarified by summarizing the topic and adding a figure to illustrate it.
The communities that received the intervention in Stage I (intervention communities) and those that did not (control communities) are not specified in the Materials and Methods section.
4) Results
The results could be improved.
Some points need to be clarified.
In p.5, lines 186-188, the authors affirm that “(…) 175 (37.1%) spoke the Mayan language and Spanish and 297 (62.9%) only understood the Mayan language but did not speak it.” In the methods, the results, and the discussion, it is not clear why it is important to speak/ understand the Mayan language. Is the Mayan language used in health communication? Is it important to understand health literacy?
In topic 3.2 Analysis of barriers
Why not show the results of the main barriers comparing the cities? Isn´t it better to prior present the main barriers (Table 8) to show then the percentage of women who had or did not have transportation for the transfer (Table 7)? Does the lack of money have a relation with the transportation barrier?
Tables 1, 2, and 3 must be improved.
In Table 1, the 19-year-old participant is out of the table, but it shows a total of 472 participants and 100% of the percentage, but the summary of the number of participants presented in the table is 471 e the percentage of 99.8%.
Table 2 presents the intervention and control communities pairs and the homogeneity between them, but it is not clear in the title.
Table 3 is very confusing. The communities' names and the results of each one are not in the same line, the authors should reorganize it. The lecture on a table must be accessible to all readers.
Table 8 presents the “Main barriers faced by mothers and caregivers when transferring for child care.” Does the “number of times mentioned” mean the number of individuals cited for each barrier type? Why not use “Number of mothers” instead of “number of times mentioned”?
5) Discussion
The organization of the health network also influences access to health (organizational barriers). Obtaining a vehicle for transportation is more critical in health access when health services are poorly distributed and far from residence. The authors could enrich the discussion by bringing in aspects of the organization of health services in the communities studied.
6) Conclusions
This section may be improved after the revisions suggested.
The authors might explain or clarify why they decided to evidence the gender conditions in conclusions once they did not approach this theme in other parts of the study.
Finally, I suggest that the educational material (illustrated booklet) and the questionnaires used be presented as supplementary material.
Comments on the Quality of English Language
The English language is appropriate.
Author Response
1) Abstract
The abstract is too extended (329 words) and not objective. Authors can reduce the introduction to the theme and focus on showing the objective, methods, main findings, and conclusion.
We have reduced the abstract.
2) Introduction. We explain briefly how Mexico's Health System is organized and the specificities of Yucatán, so that the reader from abroad can understand the study context. References added.
3) Material and methods
Repetitions are eliminated, a figure is added that better describes the activities of the previous and current intervention. We mentioned the intervened communities and control communities studied in previous intervention.
4) Results. Results are improved. It is clarified regarding the Mayan lenguage the translators and the communications on health. We have changed the order of tables 7 and 8.
Tables 1, 2, 3 and 8 are improved. We have changed according to the suggestions.
Discussion. We have done comments about the organization of health services in Yucatan.
Conclusion. The comment about the gender was made because in the communities the dicision-making power is poor. This coment is better explained in the introduction.
Educational material and questionaires will be sent.
Thank you very much for your valious comments and suggestions.
Round 2
Reviewer 1 Report
Comments and Suggestions for Authors
Dear authors:
Congratulation on improving your article.
Although it might be a relevant study regarding access to healthcare in vulnerable population, the article has still a major área of weakness, as the methodology is not throughly explained.
I also suggest a moderate editing of english language.
Best regards
Comments on the Quality of English LanguageModerate editing of English language required
Author Response
Comment 1. Although it might be a relevant study regarding access to healthcare in vulnerable population, the article has still a major área of weakness, as the methodology is not throughly explained.
Response 1. In Materials and Methods I describe with major detail the methodology carried out. I have put in red and yellow the new. I think so is more clear now. The previou paragraph (lines 169-172) could be changed with the next:
( lines 173-178) The present study is an analytic cross-sectional design (the parameters were measured only one-time) and give follow-up an educative intervention with pre post design, finalized a year before (Stage I). We utilized the same questionnaire pre post administered to the mothers and caregivers of children under five-years old, in the intervention finalized previously to the currect study. After analysis of Stage I, control communities also received the educative intervention for ethical considerations.
I thougth this because can be confuse as a follow-up study (parameters measured two o more times); but if I say: this study give follow-up an intervention study..... I think is more clear.
I would like also clear: Previous intervention the statistics analysis was with differences in differences method, with experimental and control group. After that, all people recieved the intervention. In current study, we measure only one time the variables and were compared with the last meseaurement of previous study. After of review, we think our current study is a analytical cross-sectional study design.
I have eliminated the section tituled Evaluation of previous community intervention, because the text in this section is repeated and was mentioned in previou section tituled Description of the previous community intervention (227-245) in the wich I have included new text marked in yellow.
In the section Description of the current community assessment, I have included a new paragrah initial. (248-251):
To the current study, local work team was organized again. Researchers, promoters and social workers of UADY visited to the mothers and caregivers in the 8 communities, and the same questionnaire on knowledge of alarm signs, seeking care and access to health care was applied.
(lines 259-262) text modified: …………with the support of these personnel, social workers and community promoters, the which had registered addresses of mothers and caregivers, contacted them again. After informed consent was signed, the same survey from the evaluation intervention carried out 12 months ago
(lines 277-289). In Community evaluation of current study, I say with more detail the statistics analysis. Differences in means and percentages were calculated. OR to measure association.
Limitations. I have included a new limitation (lines 475-477) and (line 480).
I hope with this have been more clear with descirption of the methodology, the which was amplied.
Comment 2. I also suggest a moderate editing of english language.
Response 2. About this, We will solicit support to MDPI
Please see the attachment

Reviewer 2 Report
Comments and Suggestions for Authors
While the authors have done extensive edits and added more explanation. My concern with the basis of the study still remains. I am not convinced with the study design and outcomes/results. The writing quality also can be further improved.
Comments on the Quality of English LanguageThe quality of the writing has improved a lot from the last version. However, it can be better.
Author Response
Comment 1. While the authors have done extensive edits and added more explanation. My concern with the basis of the study still remains. I am not convinced with the study design and outcomes/results. The writing quality also can be further improved.
Response. Current study was designed to give follow-up to a cuasi-experimental study that finalized before begining the current. All development about the basis of the study is described in Material and Methods, in Description of the previous community intervention (lines 192-225). Description about current study and its evaluation is described from line 227 to 291.
In Materials and Methods I describe with major detail the methodology carried out. I have put in red and yellow the new. I think so is more clear now. The next paragraph could be changed with the previous (line 169-172):
( lines 173-178) The present study is an analytic cross-sectional design (the parameters were measured only one-time) and give follow-up an educative intervention with pre post design, finalized a year before (Stage I). We utilized the same questionnaire pre post administered to the mothers and caregivers of children under five-years old, in the intervention finalized previously to the currect study. After analysis of Stage I, control communities also received the educative intervention for ethical considerations.
I would like be more clear with this: Previous intervention the statistics analysis was with differences in differences method, with experimental and control group. After that, all people recieved the intervention. In current study, we measure only one time the variables and were compared with the last meseaurement of previous study. After of review, we think our current study is a analytical cross-sectional study design.
I have eliminated the section tituled Evaluation of previous community intervention, because the text in this section is repeated and was mentioned in previou section tituled Description of the previous community intervention (227-245) in the wich I have included new text marked in yellow.
In the section Description of the current community assessment, I have included a new paragrah initial. (248-251).
To the current study, local work team was organized again. Researchers, promoters and social workers of UADY visited to the mothers and caregivers in the 8 communities, and the same questionnaire on knowledge of alarm signs, seeking care and access to health care was applied.
(lines 259-262) text modified: ……with the support of these personnel, social workers and community promoters, the which had registered addresses of mothers and caregivers, contacted them again. After informed consent was signed, the same survey from the evaluation intervention carried out 12 months ago
(lines 277-289). In Community evaluation of current study, I say with more detail the statistics analysis. Differences in means and percentages were calculated. OR to measure association.
Limitations. I have included a new limitation (lines 475-477) and (line 480).
I hope with this have been more clear with the methodology, the which was amplied.
Comment 2. The quality of the writing has improved a lot from the last version. However, it can be better.
Response 2. We will solicit support to MDPI to improve the quality of the writing.
Please see the attachment

Reviewer 3 Report
Comments and Suggestions for Authors
The authors have thoroughly reviewed the aspects pointed out in the 1st review, which made the article clearer to all readers.
I want to comment on the inclusion in lines 81-83 when they affirm that "(...) the man is the one who has an income and the woman does not. This situation places her at a disadvantage, because although she knows that the sick boy or girl must be cared for, she cannot go out for help, because she does not have financial resources (14)."
The conclusion of the cited reference indicates that "(....) female respondents frequently described power as relating to women’s income generation and financial independence, as well as in terms of women being listened to generally in their social relationships and by their husbands. Whilst women’s financial independence was reported to ease marital relationship tensions and supported their ability to undertake responsibilities, men still remained the authority figures in households, often in regard to health care decision-making. The study findings suggest that economic strengthening models, such as the VSLA, can have an important role in supporting women’s economic empowerment, especially as they relate to health-related decision-making and addressing unequal gender norms" (Cornish et al., 2019).
In this sense, the article cited shows not that the women do not have an income and the men do but that even with financial empowerment, there is a social construction of male domination that guarantees male authority in decision-making. So, I suggest that the authors rewrite this excerpt of the text.
Author Response
Comment 1. The authors have thoroughly reviewed the aspects pointed out in the 1st review, which made the article clearer to all readers.
I want to comment on the inclusion in lines 81-83 when they affirm that "(...) the man is the one who has an income and the woman does not. This situation places her at a disadvantage, because although she knows that the sick boy or girl must be cared for, she cannot go out for help, because she does not have financial resources (14)."
The conclusion of the cited reference indicates that "(....) female respondents frequently described power as relating to women’s income generation and financial independence, as well as in terms of women being listened to generally in their social relationships and by their husbands. Whilst women’s financial independence was reported to ease marital relationship tensions and supported their ability to undertake responsibilities, men still remained the authority figures in households, often in regard to health care decision-making. The study findings suggest that economic strengthening models, such as the VSLA, can have an important role in supporting women’s economic empowerment, especially as they relate to health-related decision-making and addressing unequal gender norms" (Cornish et al., 2019).
In this sense, the article cited shows not that the women do not have an income and the men do but that even with financial empowerment, there is a social construction of male domination that guarantees male authority in decision-making. So, I suggest that the authors rewrite this excerpt of the text.
The authors have thoroughly reviewed the aspects pointed out in the 1st review, which made the article clearer to all readers.
I want to comment on the inclusion in lines 81-83 when they affirm that "(...) the man is the one who has an income and the woman does not. This situation places her at a disadvantage, because although she knows that the sick boy or girl must be cared for, she cannot go out for help, because she does not have financial resources (14)."
The conclusion of the cited reference indicates that "(....) female respondents frequently described power as relating to women’s income generation and financial independence, as well as in terms of women being listened to generally in their social relationships and by their husbands. Whilst women’s financial independence was reported to ease marital relationship tensions and supported their ability to undertake responsibilities, men still remained the authority figures in households, often in regard to health care decision-making. The study findings suggest that economic strengthening models, such as the VSLA, can have an important role in supporting women’s economic empowerment, especially as they relate to health-related decision-making and addressing unequal gender norms" (Cornish et al., 2019).
In this sense, the article cited shows not that the women do not have an income and the men do but that even with financial empowerment, there is a social construction of male domination that guarantees male authority in decision-making. So, I suggest that the authors rewrite this excerpt of the text.
The authors have thoroughly reviewed the aspects pointed out in the 1st review, which made the article clearer to all readers.
I want to comment on the inclusion in lines 81-83 when they affirm that "(...) the man is the one who has an income and the woman does not. This situation places her at a disadvantage, because although she knows that the sick boy or girl must be cared for, she cannot go out for help, because she does not have financial resources (14)."
The conclusion of the cited reference indicates that "(....) female respondents frequently described power as relating to women’s income generation and financial independence, as well as in terms of women being listened to generally in their social relationships and by their husbands. Whilst women’s financial independence was reported to ease marital relationship tensions and supported their ability to undertake responsibilities, men still remained the authority figures in households, often in regard to health care decision-making. The study findings suggest that economic strengthening models, such as the VSLA, can have an important role in supporting women’s economic empowerment, especially as they relate to health-related decision-making and addressing unequal gender norms" (Cornish et al., 2019).
In this sense, the article cited shows not that the women do not have an income and the men do but that even with financial empowerment, there is a social construction of male domination that guarantees male authority in decision-making. So, I suggest that the authors rewrite this excerpt of the text.
The authors have thoroughly reviewed the aspects pointed out in the 1st review, which made the article clearer to all readers.
I want to comment on the inclusion in lines 81-83 when they affirm that "(...) the man is the one who has an income and the woman does not. This situation places her at a disadvantage, because although she knows that the sick boy or girl must be cared for, she cannot go out for help, because she does not have financial resources (14)."
The conclusion of the cited reference indicates that "(....) female respondents frequently described power as relating to women’s income generation and financial independence, as well as in terms of women being listened to generally in their social relationships and by their husbands. Whilst women’s financial independence was reported to ease marital relationship tensions and supported their ability to undertake responsibilities, men still remained the authority figures in households, often in regard to health care decision-making. The study findings suggest that economic strengthening models, such as the VSLA, can have an important role in supporting women’s economic empowerment, especially as they relate to health-related decision-making and addressing unequal gender norms" (Cornish et al., 2019).
In this sense, the article cited shows not that the women do not have an income and the men do but that even with financial empowerment, there is a social construction of male domination that guarantees male authority in decision-making. So, I suggest that the authors rewrite this excerpt of the text.
The authors have thoroughly reviewed the aspects pointed out in the 1st review, which made the article clearer to all readers.
I want to comment on the inclusion in lines 81-83 when they affirm that "(...) the man is the one who has an income and the woman does not. This situation places her at a disadvantage, because although she knows that the sick boy or girl must be cared for, she cannot go out for help, because she does not have financial resources (14)."
The conclusion of the cited reference indicates that "(....) female respondents frequently described power as relating to women’s income generation and financial independence, as well as in terms of women being listened to generally in their social relationships and by their husbands. Whilst women’s financial independence was reported to ease marital relationship tensions and supported their ability to undertake responsibilities, men still remained the authority figures in households, often in regard to health care decision-making. The study findings suggest that economic strengthening models, such as the VSLA, can have an important role in supporting women’s economic empowerment, especially as they relate to health-related decision-making and addressing unequal gender norms" (Cornish et al., 2019).
In this sense, the article cited shows not that the women do not have an income and the men do but that even with financial empowerment, there is a social construction of male domination that guarantees male authority in decision-making. So, I suggest that the authors rewrite this excerpt of the text.
Response 1.
We have rewrote the excerpt of the text. Lines 85-89. It is marked in red and yellow.
This could be observed during a study with women from Sierra Leone, in which they described power related to having some income, being financially independent, and being listened to in their social groups and by their husbands. But even with this, the men authority figure prevailed in decision-making at home, even in situations related to health decision-making (14).
Please see the attachment
